# Particulate organic matter as a functional soil component for persistent soil organic carbon

Kristina Witzgall [1✉], Alix Vidal [1], David I. Schubert[2], Carmen Höschen [1], Steffen A. Schweizer [1], Franz Buegger[3], Valérie Pouteau[4], Claire Chenu[4] & Carsten W. Mueller [1,5]

The largest terrestrial organic carbon pool, carbon in soils, is regulated by an intricate connection between plant carbon inputs, microbial activity, and the soil matrix. This is manifested by how microorganisms, the key players in transforming plant-derived carbon into soil organic carbon, are controlled by the physical arrangement of organic and inorganic soil particles. Here we conduct an incubation of isotopically labelled litter to study effects of soil structure on the fate of litter-derived organic matter. While microbial activity and fungal growth is enhanced in the coarser-textured soil, we show that occlusion of organic matter into aggregates and formation of organo-mineral associations occur concurrently on fresh litter surfaces regardless of soil structure. These two mechanisms—the two most prominent processes contributing to the persistence of organic matter—occur directly at plant–soil interfaces, where surfaces of litter constitute a nucleus in the build-up of soil carbon persistence. We extend the notion of plant litter, i.e., particulate organic matter, from solely an easily available and labile carbon substrate, to a functional component at which persistence of soil carbon is directly determined.

[1] Soil Science, TUM School of Life Sciences, Technical University of Munich, Freising-Weihenstephan, Germany. [2] Institute for Organic Farming, Soil and Resource Management, Bavarian State Research Center for Agriculture, Freising-Weihenstephan, Germany. [3] Institute of Biochemical Plant Pathology, Helmholtz Zentrum München (GmbH), German Research Center for Environmental Health, Neuherberg, Germany. [4] UMR Ecosys, INRA AgroParisTech, Bât. EGER, Thiverval-Grignon, France. [5] Department of Geosciences and Natural Resource Management, University of Copenhagen, Copenhagen, Denmark. ✉email: kristina.witzgall@tum.de

Sustained by a continuous input of plant-derived carbon (C), soils comprise the largest terrestrial C pool; therefore, this pool has a decisive role in the global C cycle[1,2]. Microbial decomposition is a crucial process in transforming plant-derived organic matter (OM) and in fostering the formation of soil organic matter (SOM). Consequently, the abundance and activity of microorganisms determine the pathway of C from early-stage plant litter residues to persistent SOM[3,4]. In turn, the microbiome is controlled by the soil environment, where biological, chemical and physical factors determine microbial growth and activity[5,6].

One major factor for the biogeochemical functioning of soils is the 3D arrangement of solids. The physical soil structure defines the porous network, affecting the movement and bioavailability of gases (e.g., $CO_2$ and $O_2$) and water[6]. Determined by pore size, the differences in soil water contents can shape ecological niches suitable for certain microbial taxonomic groups. The size of pores also controls the contact between microorganisms and their essential source of energy and nutrients—the litter[7]. The effect of soil structure on the functionality of the microbial community can be predicted, e.g., via oxygen availability, which regulates C turnover[8].

Over the last decades, it has become more evident that inherent chemical recalcitrance is of less importance to SOM persistence compared to soil structure-driven mechanisms that rely on soil aggregation and accessibility of reactive mineral surfaces[9,10]. Soil C is mostly stored in two major pools: as particulate OM (POM; particulate organic residues mostly of plant origin) and mineral-associated OM (MAOM; OM adhering to mineral surfaces)[11,12]. Physical mechanisms, such as the potential of OM compounds to adhere to mineral surfaces[13], or the accessibility of substrates for microorganisms[3], are now paving the way for a better understanding of OM cycling and persistence in soils. This persistence of soil C is regulated in microscale hot spots[14] at which microorganisms transform plant-derived OM into SOM[15]. The functioning of biogeochemical interfaces between plant litter substrates, microorganisms, and soil mineral surfaces requires chemical, physical, and biological factors to be considered in the consortium.

Here, we apply a systemic approach by investigating how physical soil texture governs the pathway of litter-derived C compounds from initial plant litter into more persistent SOM pools via microbial transformation in a relevant process scale (μm–mm). To disentangle mineral-microorganism interactions that regulate these processes, we incubate two differently textured soils together with $^{13}$C-labeled litter in a 95-day microcosm experiment. Aside from monitoring $CO_2$ production and litter-derived $^{13}CO_2$ release, we divide the microcosms into three depth increments and follow alterations in the chemical composition of SOM in POM and MAOM, and assess microbial community structures and microbial uptake of litter-derived $^{13}$C into phospholipid fatty acids (PLFA). The intact biogeochemical interfaces between plant residues (POM), microorganisms, and soil minerals were, for the first time, directly studied using nano-scale secondary ion mass spectrometry (NanoSIMS). Our objective in this study is to quantitatively assess interactions between microbial litter decay and the parallel formation of more persistent soil C pools in regard to aggregate formation, as well as the association of microbial C with mineral surfaces. Our data show that soil texture controls microbial activity and fungal growth, where litter addition leads to enhanced $CO_2$ release and fungal abundance in the coarser-textured soil compared to the finer-textured soil. The PLFA profiles further confirm that fungi mainly consume the litter, and transform it into microbial biomass. The expansion of the fungal hyphae leads to the build-up of litter-derived OM into deeper soil layers in the coarser-textured soil. This highlights the key role of fungi not only for litter

decomposition but also for the translocation of OM away from the detritusphere. This is further underlined by the elemental and chemical composition of POM and MAOM fractions, showing the occlusion of fresh litter-derived C into soil aggregates in deeper soil layers away from the litter layer. Via direct measurements of intact plant-soil interfaces at the nano-scale, we show the spatial distribution and isotopic enrichment of fungal hyphae, microorganisms, and soil minerals assembled at litter surfaces. We conclude that POM surfaces are not only constituting hotspots for microbial activity, but also for the occlusion and formation of mineral-associated litter-derived OM regardless of soil texture, and ultimately, for the regulation of SOM persistence.

## Results

**Litter decomposition and native soil carbon priming.** We measured how different soil textures and litter addition (enriched in $^{13}$C, $\delta^{13}$C = 2129 ± 82 ‰ V-PDB) affected the soil heterotrophic respiration by monitoring the soil-$CO_2$ emissions. By analyzing $^{13}CO_2$, we were able to differentiate $CO_2$ derived from native soil organic C from $CO_2$ derived from the added litter. In the coarse-textured soil, the total native respiration (89.4 mg $CO_2$-C $g^{-1}$ $C_{bulk}$) and the net litter-derived $CO_2$ (40.4 mg $CO_2$-C $g^{-1}$ $C_{bulk}$) were significantly higher than in the fine-textured soil (55.3 mg $CO_2$-C $g^{-1}$ $C_{bulk}$ and 24.6 mg $CO_2$-C $g^{-1}$ $C_{bulk}$, $P < 0.001$, $t = -7.512$ and $t = -6.593$ respectively; df = 8 for both, Fig. 1). While the litter-derived $CO_2$ accounted for around 30% of the total respiration in both soil textures, the litter addition induced a higher total priming effect in the coarse-textured soil (Fig. 1b and d), accounting for a net release of 23.3 mg $CO_2$-C $g^{-1}$ $C_{bulk}$ from the native

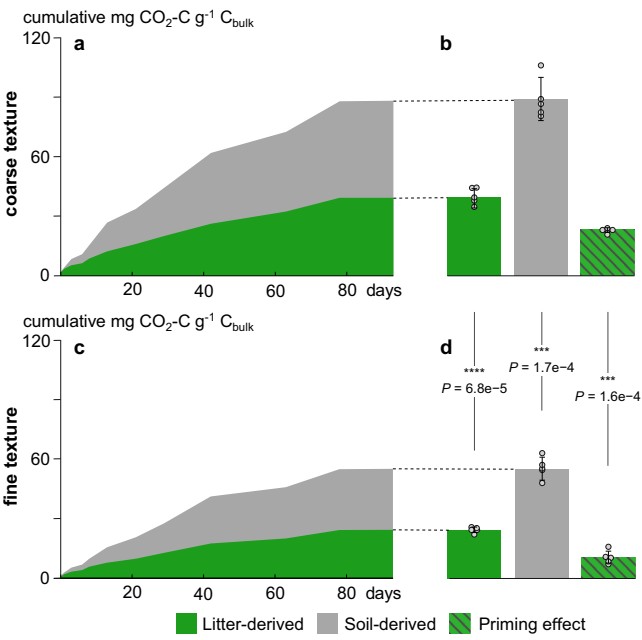

**Fig. 1 Cumulative heterotrophic respiration in fine-textured and coarse-textured soil.** Respired $CO_2$-C $g^{-1}$ $C_{bulk}$ during the 95-day incubation in (**a/b**) coarse-textured and (**c/d**) fine-textured soil. The total respired $CO_2$-C in soil with (**b**) coarser and (**d**) finer texture is displayed on the right (means, SDs displayed with errors bars, n = 5 independent replicates), together with the total priming effect. We report $CO_2$-derived C per amount C in incubated samples to directly showcase the mechanistic process level. Asterisks represent significant differences between the textures (***$P < 0.001$, ****$P < 0.0001$). Statistical significance was analyzed using an unpaired two-sided t-test.

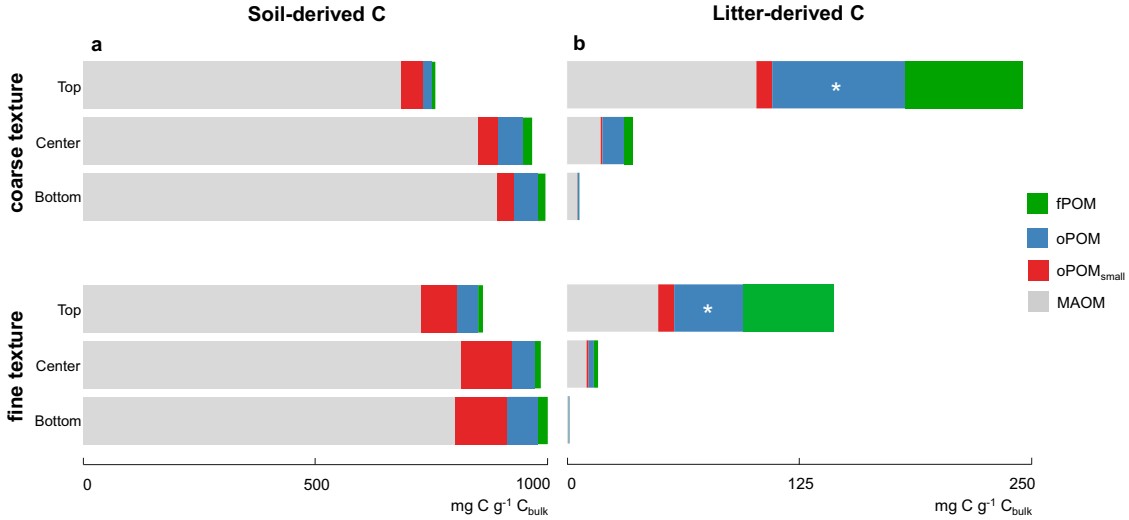

**Fig. 2 Allocation of soil-derived and litter-derived C to OM fractions.** Content of free particulate organic matter (fPOM), occluded POM (oPOM, oPOM$_{small}$), and mineral-associated OM (MAOM) in mg C g$^{-1}$ C$_{bulk}$ of (**a**) soil and (**b**) litter origin in three depths of coarse-textured and fine-textured soil (means, $n = 3$ independent replicates). Asterisks represent significant differences between the textures (*$P = 0.007$). Statistical significance was analyzed using an unpaired two-sided $t$-test.

soil organic C in the coarse-textured soil compared to 10.8 mg CO$_2$-C g$^{-1}$ C$_{bulk}$ in the fine-textured soil ($P < 0.001$, $t = -7.686$, df $= 8$).

**The fate of litter-derived carbon in particulate and mineral-associated OM fractions.** We assessed the contribution of OC derived from the decaying litter to the formation of differently stabilized OM pools in two soils with contrasting textures divided into three depths (top, center, and bottom) by soil fractionation according to density and size. The soil-derived C in mg g$^{-1}$ C$_{bulk}$ was similarly distributed across OM fractions for both differently structured soils. The MAOM fraction dominated the C storage in both soils (Fig. 2a). In the coarse-textured soil, we found a significantly higher litter-derived C content occluded within aggregates (oPOM; 71.1 mg C g$^{-1}$ C$_{bulk}$ compared to 36.8 mg C g$^{-1}$ C$_{bulk}$ in the fine-textured soil, $P = 0.007$, $t = -5.03$, df $= 4$) and slightly higher content in the MAOM fraction (101.3 mg C g$^{-1}$ C$_{bulk}$ in the soil with coarser texture compared to 48.8 mg C g$^{-1}$ C$_{bulk}$ in the soil with a finer texture, $P = 0.08$, W $= 0$; Fig. 2b). Although not statistically significant, a tendency of a higher contribution of litter-derived C recovered as oPOM and MAOM in the coarse-textured soil further extended down with soil depth to the center layer of the microcosms ($P = 0.06$ in both cases, $t = -2.66$ and $-2.60$, respectively; df $= 4$ for both).

**Fresh litter incorporated into soil aggregate structures.** The chemical composition of OM fractions was analyzed using $^{13}$C solid-state nuclear magnetic resonance spectroscopy (NMR). Carbohydrates (O/N alkyl C) clearly dominated the NMR spectra of all fPOM fractions (Fig. 3), and similar chemical composition was also detected in oPOM fractions of both textures with litter. In oPOM fractions, the added litter had induced an increase in relative intensity from around 50% to 70% in the O/N alkyl C region (Supplementary Table 1), demonstrating the dominance of polysaccharides (mainly cellulose and hemicellulose)[16]. According to the molecular mixing model results, this was further supported by the relative increase in carbohydrates (+26% in the coarse-textured and +20% in the fine-textured soil) accompanied by a relative decrease in lignin (−12% in the coarse-textured and −10% in the fine-textured soil) in oPOM fractions (Supplementary Table 2)[17,18]. The incorporation of litter-derived

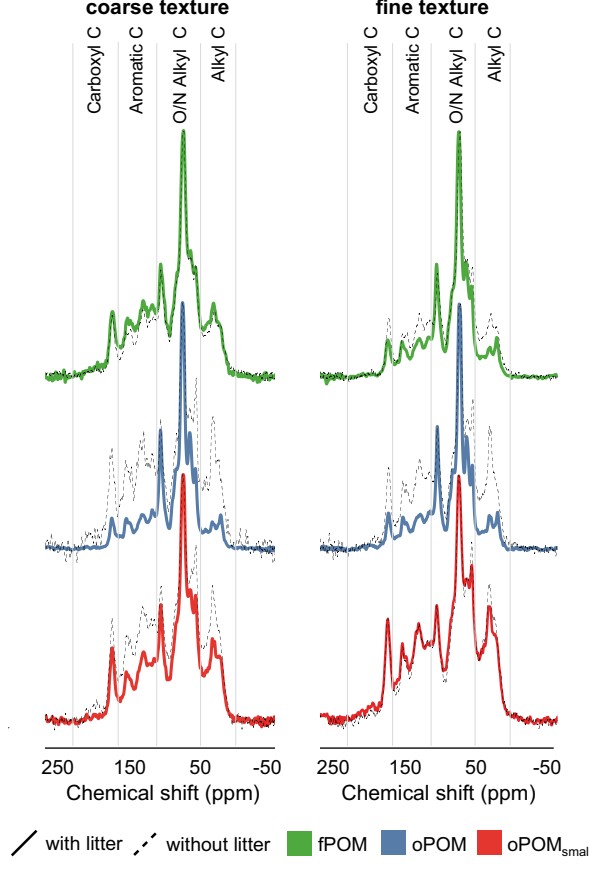

**Fig. 3 Differences in the chemical composition of particulate OM fractions.** Solid-state $^{13}$C CP-MAS NMR spectra displaying the chemical compositions of free particulate organic matter (fPOM) and occluded POM (oPOM, oPOM$_{small}$) in coarse-textured and fine-textured soil (control samples in black). The chemical shift regions represent the following functional groups: 0-45 ppm (alkyl C), 45–110 ppm (O/N alkyl C), 110–160 ppm (aromatic C), and 160–220 ppm (carboxyl C). $n = 3$ independent replicates for fPOM with litter (both textures), oPOM with litter (both textures), and oPOM$_{small}$ with litter in the finer texture. For the rest of the samples, $n = 1$.

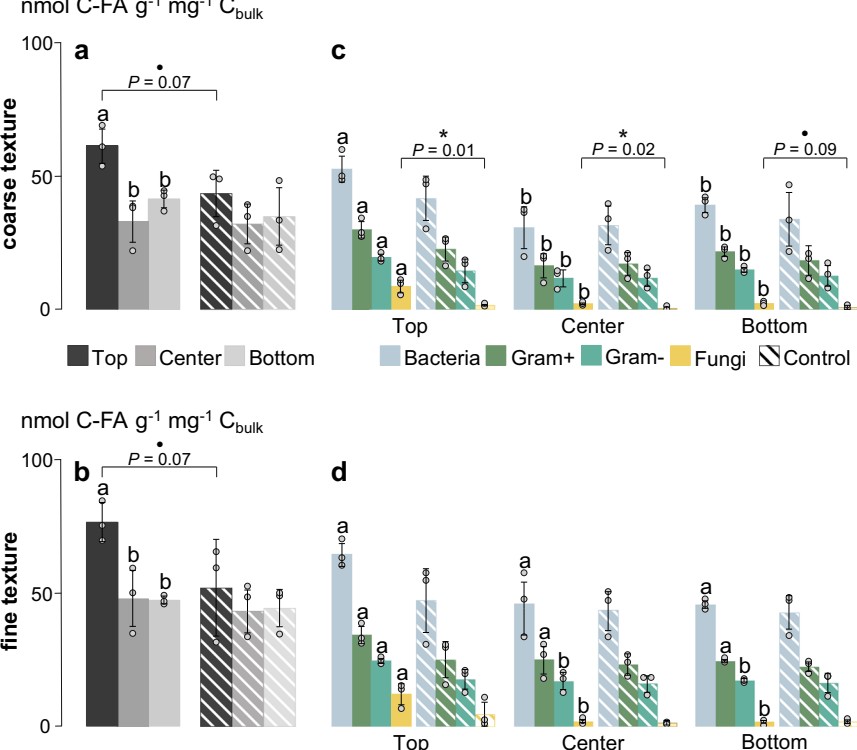

**Fig. 4 Community structures and functionality of microorganisms.** The total abundance of phospholipid fatty acids (PLFA) normalized for bulk C in nmol C-FA $g^{-1}$ $mg^{-1}$ $C_{bulk}$ (means, SDs displayed with errors bars, $n = 3$ independent replicates) in soil with (**a**) coarser and (**b**) finer texture, divided into four microbial subgroups in (**c**) coarse-textured and (**d**) fine-textured soil. Significance levels indicated by dots and asterisks (*$P < 0.1$, *$P < 0.05$) represent differences between litter treatment and controls, and lowercase letters represent significant ($P < 0.05$) differences between top, center, and bottom layers. Control samples are displayed as hatched. Statistical significance between litter treatment was analyzed using an unpaired two-sided $t$-test and differences between depths using one-way ANOVA with Tukey's HSD as the post-hoc test.

OC into soil aggregate structures was also demonstrated by the decrease in aliphaticity (alkyl C:O alkyl C ratio) in oPOM compared to control samples (Supplementary Table 1).

**Fungi respond the strongest to litter addition.** The changes caused by litter addition in microbial community structures between the textures were captured via the measurement of microbial-derived PLFA. The litter amendment led to a slight increase in the total PLFA content in the top layers of both the coarse-textured (61 nmol $g^{-1}$ mg soil $C^{-1}$, $P = 0.07$, $t = -2.40$, df = 4) and fine-textured soil (76 nmol $g^{-1}$ mg soil $C^{-1}$, $P = 0.10$, $t = -2.15$, df = 4), whereas the total PLFA contents in the center and bottom layers were similar to those of the controls (Fig. 4a and b). While the differences in soil texture had no effect on the overall community structure, a strong response to litter addition was detected in fungal biomarkers. The increase in fungal markers was particularly pronounced in the top layer of the coarse-textured soil where fungal abundance increased by a factor of 5.4 ($P = 0.01$, $t = -4.11$, df = 4) compared to 2.6 in the fine-textured soil ($P = 0.15$, $t = -1.75$, df = 4). As opposed to the other observed microbial groups, the increase in fungal markers also extended into the center layer of microcosms with coarse-textured soil (2.1 nmol $g^{-1}$ mg soil $C^{-1}$ compared to 0.4 nmol $g^{-1}$ mg soil $C^{-1}$ in the control, $P = 0.02$, $t = -3.81$, df = 4; Fig. 4c), while there was no corresponding increase in the respective layers in the finer-textured soil (Fig. 4d).

When considering the proportion of fatty acids with incorporated litter-derived $^{13}C$ within the observed groups in relation to the total amount of enriched FAs in the sample, neither texture nor depth had an effect (Fig. 5a). This corroborated the consistent community structures detected during the PLFA analysis. When

considering the proportions of $^{13}C$-enriched FAs to unlabeled FAs within each microbial group, the proportion was by far the highest in the fungal markers (92% in the coarser-textured and 82% in finer-textured soil, $P = 0.11$ between textures, $t = -2.04$, df = 4, Fig. 5b). This distinction of fungi compared to other microbial groups was significant in top layers of both textures, as well as in the center layer of the coarse-textured soil (over 42% of FAs were enriched compared to 21% in the fine-textured soil). Furthermore, the proportion of enriched gram-negative markers were significantly higher in all layers of the coarser-textured soil ($P = 0.004$, $t = -5.79$, $P = 0.019$, $t = -3.83$, and $P = 0.0007$, $t = -9.54$; df = 4 for all) compared to the fine-textured soil.

**Formation of MAOM fostered by microbial activity on decaying POM surface.** We gained direct insight into the biogeochemical interface between decaying plant residues (POM), mineral particles, and microorganisms at the microscale using scanning electron microscopy (SEM; Fig. 6, Fig. 7a1 and b1, Supplementary Fig. 1) and NanoSIMS. This approach allowed to determine elemental and isotopic information on fungal hyphae, unicellular microorganisms and extracellular polymeric substances (EPS; $^{12}C^{14}N^-$ distribution), soil minerals ($^{16}O^-$ distribution), and litter-derived POM particles ($^{12}C^-$ distribution).

We quantified the $^{13}C$-enrichment in EPS covering large areas of the POM particles, forming a biofilm-like structure that was intertwined with $^{13}C$ labeled fungal hyphae and unicellular microorganisms (Supplementary Figs. 2 and 3). Clay-sized minerals were directly enclosed into the biofilm on the POM surface (Figs. 6 and 7b). The microorganisms and EPS were significantly enriched in N compared to the underlying POM, with a

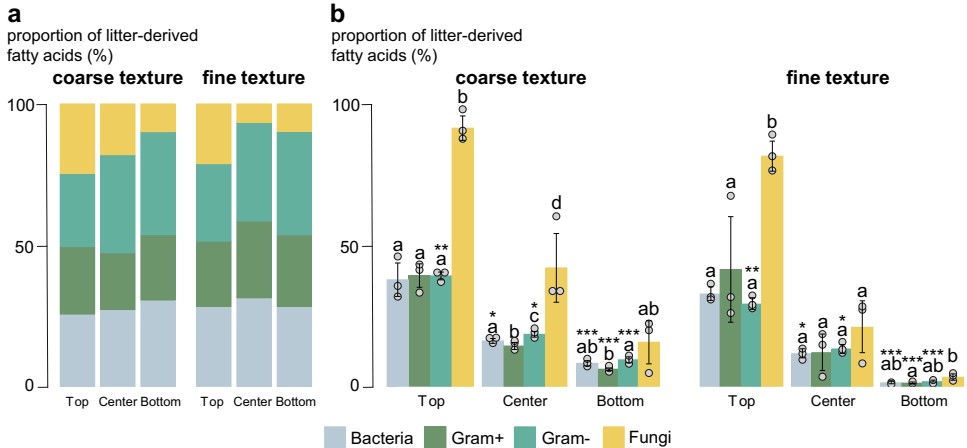

**Fig. 5 Litter incorporation in microbial biomass. a** The proportion of litter-derived fatty acids within a certain microbial group related to the total amount of litter-enriched fatty acids in the entire sample (%) in the three depths of two textures (means, $n = 3$ independent replicates). **b** The proportion of $^{13}C$-enriched fatty acids compared to unlabeled fatty acids within each microbial group (%) in the three depths of two textures (means, SDs displayed with errors bars, $n = 3$ independent replicates). Lowercase letters represent significant differences ($P < 0.05$) between microbial groups and asterisks (*$P < 0.05$, **$P < 0.01$, ***$P < 0.001$) represent differences between textures. Statistical significance between texture was analyzed using an unpaired two-sided $t$-test and differences between microbial groups with the Kruskal-Wallis test. Differences between depths were significant in all groups ($P < 0.05$; one-way ANOVA with Tukey's HSD).

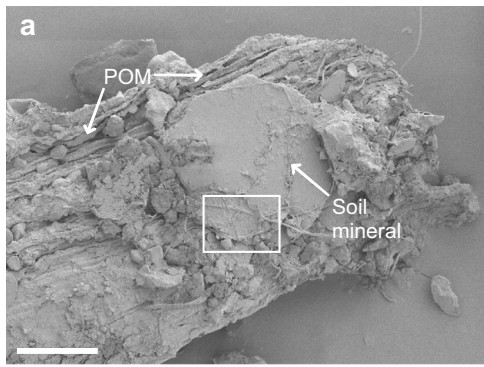

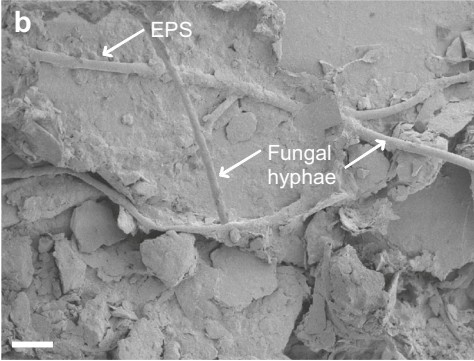

**Fig. 6 SEM micrographs of the spatial connection of organo-mineral assemblages.** Scanning electron microscopy (SEM) images of the interface of plant litter (POM) and soil minerals, where soil minerals are (**a**) attached to the litter surface (scale bar = 100 μm) and (**b**) enmeshed with fungal hyphae and extracellular polymeric substances (EPS; scale bar = 10 μm). Similar images were obtained from at least 10 independent locations in each soil texture.

higher $^{12}C^{14}N^-$:($^{12}C^- + ^{12}C^{14}N^-$) ratio obtained for EPS, followed by hyphae and bacteria (Fig. 7d). The $^{13}C^-$:($^{12}C^- + ^{13}C^-$) ratios for fungal hyphae, bacteria, and EPS (3.0, 2.3, and 3.0 atom % $^{13}C$, respectively) were well over the natural abundance level (1.1 atom %

$^{13}C$), and the hyphae showed a significantly higher enrichment compared to bacteria and POM ($P < 0.05$, df = 3, Fig. 7c).

## Discussion

Soil texture, and thus the 3D structure of soils, controls overall microbial activity; the coarser soil texture entailed both higher decomposition of litter-derived OM, and an increased priming effect, fostering the mineralization of native soil C (Fig. 1b and d). Plant litter fragments located in larger soil pores of coarse-textured soils are more easily accessible; therefore, litter decomposition is enhanced[19–21]. In the coarse-textured soil, the increased accessibility and, hence, increased bioavailability of litter-derived C was further demonstrated by consistently higher proportions of labeled PLFAs in gram-negative bacteria across all soil depths (Fig. 5b); bacteria that are specialized in the processing of labile plant C sources[22,23].

We show that the coarse-textured soil offered a more favorable habitat for fungi in a microenvironment rich in bioavailable substrates formed by fresh unprotected litter. Fungal abundance increased by more than five-fold following the litter addition in the coarse-textured soil. Furthermore, a substantial part (92% in the coarse-textured soil) of the fungal biomass was directly derived from the added plant litter, as demonstrated by the PLFA measurements (labeled PLFA profiles; Fig. 5b). This highlights the key role of the fungal community for rapid litter decomposition, particularly in coarse-textured soils. Soil structure with a distinct soil pore network determines the abundance and community structure of microbiota[24] and fungi are mainly found in macro-pore spaces (>10 μm) that are noticeably larger than the hyphae itself[19,25,26]. The filamentous growth of the mycelium enables fungi to bridge air-filled pore spaces, supporting them to overcome capillary boundaries between wet and dry soil and to adapt to heterogeneous pore networks[27–29]. Consequently, under the physical conditions of coarse-textured soils, fungi have a clear advantage over other microorganisms to reach OM in hard-to-access soil compartments that are not connected via water nor biofilms[24]. Our microcosm experiment suggests that in sandy soils, fungi are key to sustain crucial soil functions such as C and nutrient cycling by the transformation of litter-derived OM into SOM.

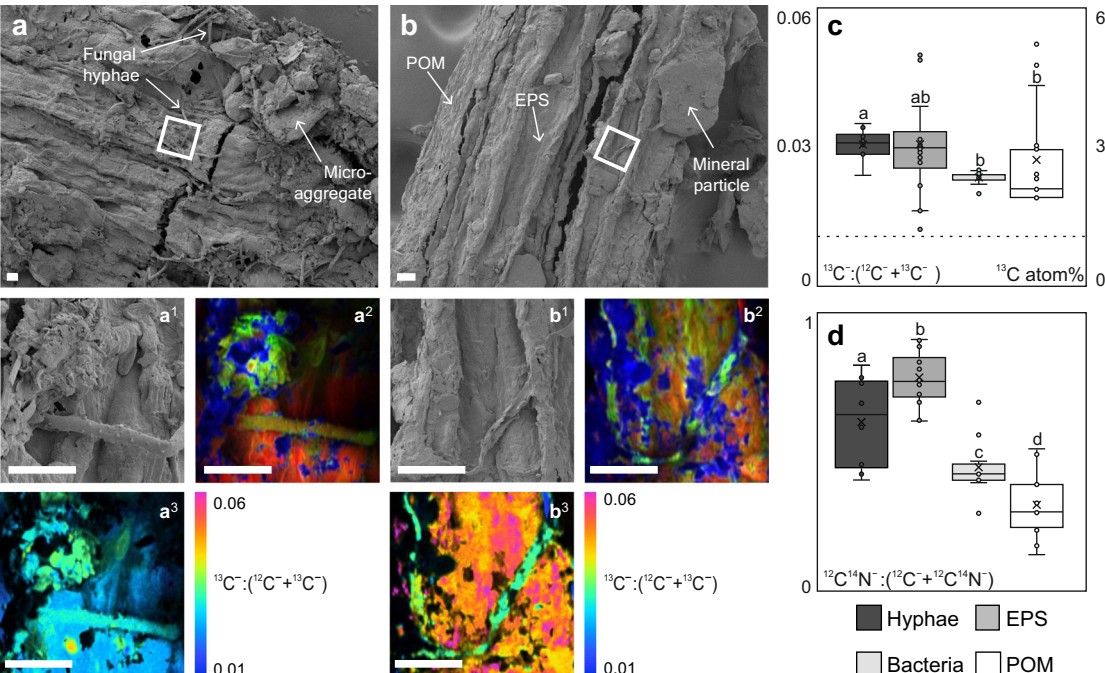

**Fig. 7 High $^{13}$C enrichment detected in fungal hyphae and EPS based on NanoSIMS imaging.** Scanning electron microscopy (SEM) images of $^{13}$C-enriched maize litter incubated in microcosms and isolated as particulate organic matter (POM) in (**a**) coarse-textured and (**b**) fine-textured soil. Similar images were obtained from at least 10 independent locations in each soil texture. **a$^1$/b$^1$** SEM micrographs of measurement spots in (**a$^1$**) coarse-textured and (**b$^1$**) fine-textured soil, which was later analyzed by nano-scale secondary ion mass spectrometry (NanoSIMS). In each soil texture, at least five NanoSIMS measurements were conducted at independent locations on the POM fragments with similar results. **a$^2$/b$^2$** NanoSIMS composite images displayed as RGB (Red = $^{12}$C$^-$, Green = $^{12}$C$^{14}$N$^-$ and Blue = $^{16}$O$^-$). **a$^3$/b$^3$** NanoSIMS hue-saturation intensity (HIS) images displaying the $^{13}$C$^-$:($^{12}$C + $^{13}$C$^-$) isotope ratios of POM, fungal hyphae, and extracellular polymeric substances (EPS) in the coarse-textured and fine-textured soil. Here the enrichment level is displayed as HIS images with a color scale ranging from natural abundance (0.01) in blue to high enrichment (0.06) in purple. Scale bars = 10 μm. **c** Boxplots of $^{13}$C$^-$:($^{12}$C$^-$ + $^{13}$C$^-$) isotope ratios and **d** $^{12}$C$^{14}$N$^-$:($^{12}$C$^-$ + $^{12}$C$^{14}$N$^-$) ratios of hyphae (n = 8), EPS (n = 16), bacteria (n = 10), and POM (n = 13) in both textures obtained by NanoSIMS. The regions of interest were selected manually on continuous fragments of hyphae, individual bacteria, patches of EPS, and exposed POM surfaces. Box plots indicate medians (line) and means (x), where the first (Q1) and third (Q3) quartile are represented by the lower, respectively upper bounds of the box. Error bars represent the data range, bounded to 1.5 × (Q3-Q1) and individual data points are noted as dots. The natural abundance of $^{13}$C is indicated by the hatched line. Significant differences (P < 0.05) between the four groups are indicated by lowercase letters.

In the coarse-textured soil, fungal activity extended away from the litter source, thereby promoting a downward transfer of litter-derived C into deeper soil layers (PLFA depth profiles; Fig. 4). This pattern can partly be attributed to the apical properties of the fungal mycelium, enabling the translocation of C sources throughout the fungal colony[30–32]. The expansion of hyphal networks facilitates the incorporation of litter into aggregates[33,34]. Microorganisms have a decisive role in soil aggregate formation processes[35], and the stabilization of aggregated soil structures can partly be ascribed to the exudation of EPS (e.g., polysaccharides, Fig. 7a and b) from the hypha[36]. We propose that the expansion of fungal hyphae, together with its interactions with mineral particles, results in the build-up of litter-derived oPOM in the deeper soil layers away from the litter source (Fig. 2b). This intricate interaction between fungal hyphae, plant residues, and mineral particles adhering to microbial-derived EPS was underlined by spectromicroscopic imaging (Figs. 6 and 7). With the direct measurement of intact plant-fungi interfaces, we emphasize the key role of fungi in the translocation of litter-derived C within soils, as well as in the formation of aggregates and mineral-associated OM–a process which ultimately drives the stabilization of litter-derived C compounds in soils[37].

We were able to demonstrate the incorporation of plant C into microbial biomass directly at the interface of plant residues and soil minerals. This was quantified with high levels of $^{13}$C enrichment in fungal hyphae and microbial EPS on the POM surface (Fig. 7). The direct contact between minerals ($^{16}$O$^-$ distribution; Fig. 6) and microbial biomass ($^{12}$C$^{14}$N$^-$; Fig. 7), together with the enmeshment of fresh litter (POM) with fungal hyphae and microbial-derived EPS (Fig. 7a and b), promotes the cohesion of fine-sized soil minerals[38,39]. This agglomeration of fine mineral particles, driven by microbial activity and regulated by the bioavailability of litter-derived C, drives the aggregate formation and soil structure development[40] directly at the plant-soil interface. In addition, the chemical composition of the litter-derived OM that got entrapped in soil aggregates (oPOM) by this soil structure formation resembled the undecomposed litter (Fig. 3). Thus, particulate OM acts as an important precursor for the aggregate formation[41] and parallel occlusion of litter-derived POM into aggregated soil structures (Fig. 8).

Regardless of soil texture, fresh litter surfaces serve as hotspots of microbial activity driving the formation of organo-mineral associations in concert with comprising a nucleus for aggregate formation. Thus, the biogeochemical interfaces of decaying plant litter determine—via promoted microbial activity—the two most prominent mechanisms which increase the persistence of OC in soils; the (i) occlusion of POM in soil aggregates and (ii) the association of OM with mineral surfaces as simultaneous processes across soils of different structure (Fig. 8). These two mechanisms strongly rely on the spatial proximity of particulate litter and its surfaces, microbial residues, and fine-sized mineral particles. Thus, the formation of persistent POM and MAOM,

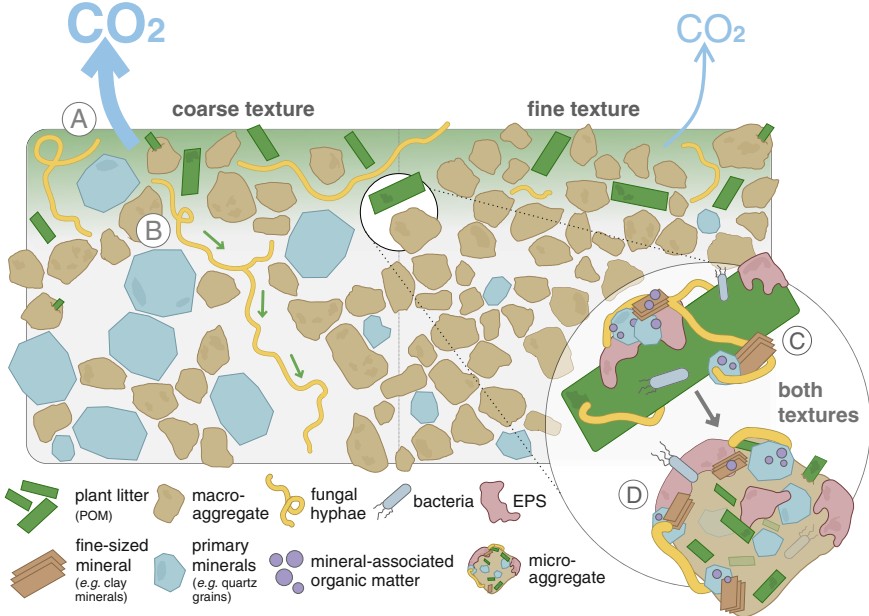

**Fig. 8 Aggregate and mineral-associated organic matter formation in soils of different textures driven by interactions between litter, microorganisms, and soil matrix.** Fresh litter surfaces serve as hotspots of microbial activity driving the formation of organo-mineral associations in concert with comprising a nucleus for aggregate formation. **A** Coarse soil texture fosters higher mineralization of native and litter-derived organic matter resulting in higher $CO_2$ emissions compared to the fine-textured soil. **B** Fungal hyphae in coarse-textured soils promote the translocation of litter-derived C away from the litter source. **C** Regardless of texture, gluing of fine-sized minerals, driven by microbial products (EPS) on the fresh litter surface leads to (**D**) the formation of soil aggregates directly at the plant-soil interface.

both constituting soil organic C pools with low turnover times[4,11], is directly fueled by the decomposition of POM and controlled by microbial activity.

## Methods

**Study site and soil sampling**. The soil was collected at $5 - 20$ cm (Ap horizon) from an agricultural field in Southern Germany (Freising, Bavaria, 48°23'53.8"N, 11°38'39.7"E) in December 2017. The sampling area is situated within the lower Bavarian upland, and characterized by a mean annual temperature of 7.8 °C and mean annual precipitation of 786 mm. The soil type is a Cambisol (silty clay loam; 32% clay, 53% silt, and 14% sand) with a considerable amount of loess mixed with underlying Neogene sandy sediments. The soil was selected to represent a widely distributed soil type and land use. The collected soil was oven-dried (2 days, 40 °C), sieved (<2 mm), and visible plant remains were manually removed using tweezers.

**Experimental setup**. The experimental design involved four treatments; soils of two textures, either with or without $^{13}C$-labeled maize stalks. In order to obtain a coarse-textured soil (sandy clay loam; 24% clay, 15% silt, and 60% sand), half of the initial soil was mixed with cleaned quartz sand (Quarzwerke, Frechen, Germany) to increase the sand content from 14% to 60% (Supplementary Table 3). To achieve consistent bulk densities between treatments (0.9–1.3 g cm$^{-3}$) 120 g (for coarser texture) and 90 g (for finer texture) soil was filled homogeneously and gently packed into microcosms (height: 5 cm, internal diameter: 5 cm, total volume: 98.2 cm$^3$; polyoxymethylene, 1.4 g cm$^{-3}$; Sahlberg, Munich, Germany). While the control microcosms were filled entirely with soil, about 330 mg of air-dried and grounded $^{13}C$-labeled maize stalks (2–3 mm, δ$^{13}C$ = 2129 ± 82‰ V-PDB; Supplementary Table 4; Agroscope, Zurich, Switzerland) were mixed into the upper 1.67 cm of the soil within the other microcosms to create a quasi-natural gradient, with above-ground litter addition from the top (Fig. 9). Maize was chosen as litter substrate as it is a crop grown worldwide in agricultural systems. Each of the four treatments was replicated five times. The microcosms were sealed from below with polyester gauzes (37-μm mesh) and placed into Ball Mason Jars (475 ml) on top of metal grids to ensure downward gas diffusion.

**Heterotrophic respiration**. After making all containers gas-tight and rinsing them with synthetic air (Westfalen AG, Münster, Germany), 12 ml of gas samples (IVA Analysentechnik, Meerbusch, Germany) were collected from the headspace of the Mason Jars on day 2, 3, 4, 8, 10, 15, 23, 31, 44, 65, 80, and 95. For each measurement of $CO_2$ respiration, two samplings of the container atmosphere were carried out, and the time in between the two samplings was adapted to the current respiration rates. During the incubation period of 95 d, the $CO_2$ concentration, as

well as the $^{13}C$ abundance in the respired $CO_2$, was measured via gas chromatography isotope ratio mass spectrometry (GC/IRMS; Delta Plus, Thermo Fisher, Dreieich, Germany). The $CO_2$ levels were calibrated against three calibration gases (890, 1500, and 3000 ppm $CO_2$; Linde AG, Pullach, Germany). Then, $CO_2$ with known isotopic composition, diluted in helium, was used as a lab standard. This standard was in turn calibrated against three international standards (RM 8562, RM 8563, and RM 8564; International Atomic Energy Agency, Vienna, Austria) with a dual inlet system. The temperature and water holding capacity were kept constant at 21 °C and 60%, respectively, along with the incubation period.

**Sampling**. Following 95 days of incubation, each microcosm was cut into three horizontal sections with a razor blade, separating the top, center, and bottom layer (each 1.67-cm high). The microcosms were designed to be opened from the side, allowing for precise separation of the depth increments (Supplementary Fig. 4). Subsamples for subsequent microbial analyses were freeze-dried and stored at 4 °C, and dried aliquots for fractionation were stored in sealed plastic containers at 20 °C. Furthermore, a few POM particles were selected manually for NanoSIMS measurements.

**Physical fractionation and subsequent analyses**. The soil was separated into five distinct OM fractions using a combined density and particle size fractionation scheme[11]. Air-dried soil (18–20 g) was gently capillary-saturated with sodium polytungstate solution (Na$_6$[H$_2$W$_{12}$O$_{40}$]; 1.8 g cm$^{-3}$) and after 12 h, the free-floating particulate organic matter (fPOM) was collected using a vacuum pump. oPOM was released from aggregated soil structures via ultrasonic dispersion (Bandelin, Sonoplus HD 2200; energy input of 440 J ml$^{-1}$)[42] allowing its separation from heavier minerals. The excess salt was removed from the oPOM by washing it with deionized water over a sieve (20-μm mesh size), which yielded an oPOM fraction of <20 μm (oPOM$_{small}$). Both fPOM and oPOM fractions were washed several times using deionized water and pressure filtration (20-μm mesh) until the solution dropped below an electric conductivity of <5 μS cm$^{-1}$ via pressure filtration. The oPOM$_{small}$ fraction was cleaned via saturation with deionized water for 24 h. While sand and coarse silt fractions were separated by wet sieving, mineral fractions <20 μm were separated via sedimentation, and later combined as one MAOM fraction. The C, N, and $^{13}C$ contents were determined for freeze-dried and milled OM fractions, as well as milled bulk soil, via dry combustion with an isotope ratio mass spectrometer (Delta V Advantage, Thermo Fisher, Dreieich, Germany) coupled with an elemental analyzer (Euro EA, Eurovector, Milano, Italy). Acetanilide was used as a lab standard for calibration and to determine the isotope linearity of the system, and was in turn calibrated against several suitable isotope standards (International Atomic Energy Agency, Vienna, Austria). International and lab isotope standards were included in every sequence to create a final $^{13}C$

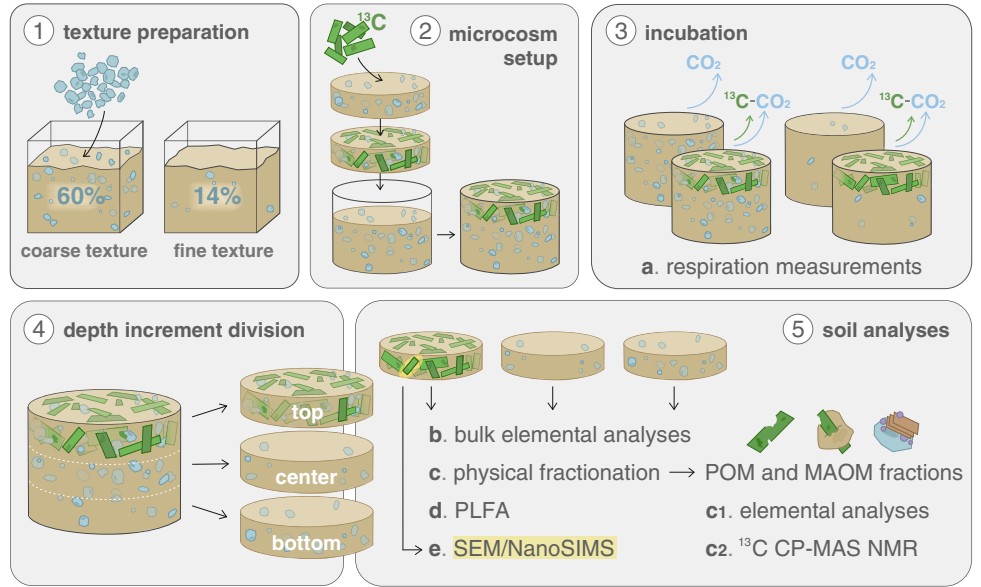

**Fig. 9 Experimental setup of $^{13}$C labeled litter incubation in soils with different textures.** (**1**) The coarse soil texture was obtained by adding quartz sand to the loamy soil material, achieving a total sand content of 60% compared to 14% in the finer-textured soil consisting only of the original soil material. (**2**) The uppermost layer of the microcosm was prepared by mixing $^{13}$C labeled plant litter with homogenized soil. Two-thirds of the microcosms were filled with soil material, and the prepared litter layer was added as the top layer in respective textures. (**3**) Five replicates for each of the four treatments (two soil textures with or without plant litter added) were incubated for 95 days. Respiration measurements (**a**) were carried out on all five replicates. (**4**) Each microcosm was cut into three depths and depth increments were thereafter always analyzed separately. (**5**) While bulk analyses (**b**) were conducted on all five replicates, three representative replicates were selected based on C and N concentrations for physical fractionation (**c**) and phospholipid fatty acid analysis (PLFA; **d**). Obtained organic matter fractions (POM and MAOM) were analyzed for C, N, and $^{13}$C (**c1**) and the chemical composition of the fractions was determined via $^{13}$C CP-MAS NMR spectroscopy (**c2**). Lastly, scanning electron microscopy (SEM) and nano-scale secondary ion mass spectrometry (NanoSIMS) measurements (**e**) were carried out on various fragments of particulate OM that had been handpicked from samples of both textures.

correction. Since the samples did not contain carbonates, the C contents were assumed to be equal to organic C contents.

**$^{13}$C nuclear magnetic resonance spectroscopy**. The chemical compositions of the POM fractions were determined via $^{13}$C CP-MAS NMR in solid-state (Bruker DSX 200, Bruker BioSpin GmbH, Karlsruhe, Germany), where samples were filled into 7-mm zirconium dioxide rotors and spun in a magic angle spinning probe at a rotation speed of 6.8 kHz and 0.01024 s acquisition time. The recorded $^{13}$C spectra were quantified in the following chemical shift regions: alkyl C ($-10$–$45$ ppm), O/N alkyl C (45–110 ppm), aromatic C (110–160 ppm), and carbonyl/carboxyl C (160–220 ppm)[11]. The regions were integrated and an alkyl C:O alkyl C ratio ($-10$–$45/45$–$110$ ppm) was computed to describe the degree of aliphaticity of the different fractions[43]. Lastly, the obtained spectra were transformed into OM compound classes via the molecular mixing model[18,44] with the following chemical shift regions: 0–45, 45–60, 60–95, 95–110, 100–145, 145–165, and 165–215 ppm.

**Calculations of litter-derived C in CO$_2$, soil, and OM fractions**. Along with the incubation period, the amount of C respired per hour was computed (Eq. 1).

$$\frac{\text{mg CO}_2\text{-C}}{h} = \frac{\triangle\text{CO}_2}{\triangle t}\left[\frac{\text{ppm}}{\text{min}}\right] \bullet \frac{1}{10^6} \bullet \frac{V_{\text{HSP}}[\text{ml}]}{24.1\left[\frac{\text{ml}}{\text{mmol}}\right]} \bullet 12\left[\frac{\text{mg CO}_2\text{-C}}{\text{mmol}}\right] \bullet 60 \text{ min} \quad (1)$$

where $\Delta\text{CO}_2/\Delta t$ is $CO_2$ increase over time, $V_{\text{HSP}}$ is the volume of the headspace of Mason Jars, the volume of an ideal gas at 21 °C is set at 24.1, and 12 represents the atomic mass of C.

Subsequently, the percentage of respired $CO_2$ originating from the litter was calculated (Eq. 2).

$$\text{CO}_2\text{-C}_{\text{litter}}[\%] = \left(\frac{\delta^{13}\text{C}_{\text{resp}} - \delta^{13}\text{C}_{\text{control}}}{\delta^{13}\text{C}_{\text{litter}} - \delta^{13}\text{C}_{\text{control}}}\right) \bullet 100 \quad (2)$$

where $\delta^{13}\text{C}_{\text{resp}}$ emission gives the $\delta^{13}$C for the current $CO_2$ emission between the two samplings (‰ V-PDB), $\delta^{13}\text{C}_{\text{control}}$ is the average $\delta^{13}$C of the control soils at the time of measurement, and $\delta^{13}\text{C}_{\text{litter}}$ is the $\delta^{13}$C signature of the labeled litter. Finally, the respired C originating from the soil was computed (Eq. 3).

$$\text{CO}_2\text{-C}_{\text{soil}}[\%] = 100 - \text{CO}_2\text{-C}_{\text{litter}} \quad (3)$$

The proportion of litter-derived C (%) in the OM fractions was calculated (Eq. 4)[45].

$$\text{Litter-derived C}[\%] = \frac{\delta^{13}\text{C}_{\text{labeled}} - \delta^{13}\text{C}_{\text{control}}}{\delta^{13}\text{C}_{\text{litter}} - \delta^{13}\text{C}_{\text{control}}} \bullet 100 \quad (4)$$

where $\delta^{13}\text{C}_{\text{labeled}}$ is the $^{13}$C enrichment in labeled samples, $\delta^{13}\text{C}_{\text{control}}$ is the $^{13}$C enrichment in controls (natural abundance level, i.e., 28‰ V-PDB), and $\delta^{13}\text{C}_{\text{litter}}$ is the $^{13}$C enrichment in the added litter (i.e., 2129‰ V-PDB) from which the amount of litter-derived C within each OM fraction could then be determined (Eq. 5).

$$\text{C}_{\text{litter}}[\text{mg}] = \frac{\text{litter-derived C}}{100} \times \text{C}_{\text{fraction}} \times m \quad (5)$$

where $\text{C}_{\text{fraction}}$ is the amount of C in mg g$^{-1}$, and $m$ is the recovered mass (g) of each fraction after the fractionation.

**PLFA analyses**. The PLFA patterns were analyzed[46] and adjusted according to the ISO/TS 29843-2:2011F standard. In summary, the soil lipids from 3 g of soil (freeze-dried aliquots) were extracted with a Bligh and Dyer solution [methanol, chloroform, and citrate buffer (pH = 4 ± 0.1), 2:1:0.8, v/v/v]. A biphasic system was achieved by adding chloroform and citrate buffer from which the lipid phase was evaporated at 30 °C under a nitrogen stream. The phospholipids were separated from neutral lipids and glycolipids by solid-phase extraction on silica tubes (SPE DSC-Si, 500 mg, Discovery®) and evaporated. The PLFA were turned into fatty acid methyl esters (FAMEs) via alkaline methanolysis[47] and later quantified via gas chromatic retention time comparison with a gas chromatograph (GC Agilent HP6890, G1530A, Chemstation, Santa Clara, USA) connected to a flame ionization detector equipped with a capillary column (SGE, BPX5, 60 m × 0,25 mm × 0,25 mm). The FAME concentrations were quantified relative to methyl non-adecanoate (19:0), enabling methylated lipids to be identified. A standard soil was used and extracted in parallel to detect potential deviations between the extraction rounds, expressed in nmol C-FA per g of soil. Mono-unsaturated and cyclopro-pylated PLFA (C16:1w7c, C18:1w9c, and C18:1w9t) were assigned to gram-negative bacteria, iso-branched and anteiso-branched PLFA (iC15:0, aC15:0, iC16:0, i-C17:0, C:17, and C18:0) were assigned to gram-positive bacteria and C18:2w6c, C18:3w3c, respectively C20:5w3c were assigned to fungi[48]. The total content of bacteria was expressed by adding gram-positive, gram-negative together with the markers C14:0, C16:0, C20:0, and C15:1. Lastly, the $^{13}$C-labeling of FAME was concluded by correcting for the added methyl moieties during methanolysis

and relating it to the chain length of fatty acids (Eq. 6).

$$\delta^{13}C_{FA}[\text{V-PDB}] = \frac{(C_n + 1) \times \delta^{13}C_{FAME} - \delta^{13}C_{MeOH}}{C_n} \quad (6)$$

where $\delta^{13}C_{FA}$ represents the $\delta^{13}C$ of the fatty acid, $C_n$ the number of C atoms in the fatty acid, $\delta^{13}C_{FAME}$ is the $\delta^{13}C$ of the fatty acid methyl ester, and $\delta^{13}C_{MeOH}$ is the $\delta^{13}C$ of the methanol used for the methylation ($-63\%$) to calculate the isotope ratios of the fatty acids. The relative incorporation of $^{13}C$ into four microbial groups was calculated by relating the proportions of each fatty acid to the total $^{13}C$ incorporation, and the absolute incorporation of $^{13}C$ in each microbial group was calculated by dividing the amount of $^{13}C$ enriched fatty acid by the total amount of extracted fatty acid for that particular group.

**SEM and NanoSIMS microspectroscopy**. In order to gain insights into the microscale distribution of the assemblages of litter with microbes and minerals, we used SEM and NanoSIMS. Free POM from non-fractionated soil was hand-picked and fixed onto graphene sample substrates on metal stubs (10 mm in diameter). To avoid the charging phenomena, samples were gold-coated prior to SEM analyses by physical vapor deposition under an argon atmosphere (Emitech Sputtercoater SC7620, Gala Instrumente, Bad Schwalbach, Germany). To analyze the microscale structures of the assemblages of POM, microorganisms and soil minerals of the samples were first analyzed using SEM (Jeol JSM 5900LV, Freising, Germany), and subsequently, the spots that best exemplified the microbial transformation on the decaying litter (POM) surface were analyzed using a Cameca NanoSIMS 50 L (Cameca, Gennevilliers, France)[49]. For the NanoSIMS measurements, a 270-pA high primary beam was used to locally sputter away impurities and gold coating, and to implant primary ions (Cs$^+$) into the sample's surface (impact energy of 16 keV) to enhance the yields of secondary ions. Subsequently, secondary ions were measured using electron multipliers; $^{12}C^-$, $^{13}C^-$, $^{12}C^{14}N^-$ to display OM fragments and $^{16}O^-$, $^{28}Si^-$, $^{27}Al^{16}O^-$, and $^{56}Fe^{16}O^-$ secondary ions to record the mineral phase. The instrument was tuned to a high mass resolution in order to accurately separate mass isobars at mass 13 ($^{13}C^-$, $^{12}C^1H^-$). The ion images were acquired with a $25 \times 25\ \mu m$ field of view, 40 planes and 1 ms pixel$^{-1}$ dwell time for all measurements. Charging effects were compensated for with an electron flood gun if necessary. The acquired measurements were dead time (44 ns) and drift corrected using the OpenMIMS plugin of the ImageJ software. The $^{13}C^-$:($^{12}C^- + ^{13}C^-$) and $^{12}C^{14}N^-$:($^{12}C^- + ^{12}C^{14}N^-$) ratios were computed for distinct regions of interests which were chosen manually with respect to the major compartments: continuous fragments of fungal hyphae, individual bacteria, EPS patches, as well as exposed POM surfaces. To account for instrumental mass fractionation, the electron multipliers were carefully checked, and the control measurements of non-labeled POM samples were conducted regularly along with the sessions. Here, the mean $^{13}C^-$:($^{12}C^- + ^{13}C^-$) ratios were in line with the level of natural abundance, which meant that a correction of ratios for labeled POM samples was not necessary.

**Statistical analyses**. All parameters were separately tested for normality with Shapiro–Wilk test and for homoscedasticity with Bartlett's test. In addition, the distribution of the datasets was checked with Q–Q plots. In cases where the assumptions of normality or homoscedasticity were not met, a log-transformation was applied on the raw data, and analyses were carried out on the log-transformed data. The differences caused by texture and litter addition were tested using unpaired $t$-tests, and depth differences were tested using one-way analysis of variance with Tukey's honestly significant difference as the post-hoc test. In cases where the log-transformed data did not meet the requirements for parametrical testing, the unpaired two-samples Wilcoxon test or Kruskal-Wallis test was applied. The statistical findings were considered significant if the confidence limits were in excess of 95% ($P < 0.05$). All statistical testing was carried out in the R statistical environment[50] using agricolae[51] and ggpubr[52] packages.

**Reporting summary**. Further information on research design is available in the Nature Research Reporting Summary linked to this article.

## Data availability
The data supporting the findings of this study are available on request from the corresponding author (K.W.).

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

## Acknowledgements

The authors gratefully acknowledge the support by the Deutsche Forschungsgemeinschaft (DFG) through grant no. HO 5121/1-1. We are thankful to Gertraud Harrington and Johann Lugmeier for their technical assistance related to the NanoSIMS measurements, Maria Greiner for the laboratory assistance, and Isabel Prater for support with the $^{13}$C CP-MAS NMR Spectroscopy measurement and evaluation. Further, we thank Juliane Hirthe for providing the $^{13}$C-labeled maize litter and Josef Reischenbeck for preparing equipment for the incubation. The authors also acknowledge Dominik Fiedler from Fraunhofer Society for the support of the SEM analyses. We are grateful for the financial support of BayFra (Franco-Bavarian University cooperation center; FK13-2018), as well as the DFG funding received for the NanoSIMS instrument [KO 1035/38-1].

## Author contributions

K.W. carried out the measurements following the incubation, collected and analyzed data, and wrote the manuscript; A.V. designed and supervised the experiment and wrote the manuscript; D.I.S. designed and conducted the incubation and respiration measurements, prepared samples for subsequent analyses, and collected and analyzed data; C.H. conducted the NanoSIMS measurements and supported the data evaluation; S.S. designed and supported the incubation experiment and contributed to the data evaluation of the NanoSIMS measurements; F.B. conducted and evaluated the GC-IRMS measurements of $CO_2$ and the EA-IRMS measurements of soil fractions; V.P. supervised the PLFA and PLFA-SIP extraction, conducted the GC-C-IRMS measurements and evaluated the data; C.C. supervised the PLFA and PLFA-SIP extraction and evaluated the data; C.W.M. designed and supervised the experiment and wrote the manuscript. All authors discussed the data and contributed to the final draft.

## Funding

## Competing interests

The authors declare no competing interests.
