## [Peer Review File · Nature Communications]

REVIEWER COMMENTS

Reviewer #1 (Remarks to the Author):

The manuscript by Witzgall et al. examines how leaf litter is ultimately processed and is incorporated into soil organic matter. I found the research here to be very interesting and is the first to my knowledge to visually observe the process of SOM formation. I believe that this will be of interest to a wide range of microbial and soil ecologists. The major issue that I have with this manuscript is that only one soil and one litter type was used in this experiment. I believe that the authors need to discuss the potential caveats that this limitation might impose. For instance how might litter quality influence the observed processes? Could variation in the soil microbial community (i.e. different soils) influence this process? As it currently stands the conclusions from this study rest on the assumption that all soils will be similar to that of a single ag soil decomposing corn litter. Additionally, a minor point, but how was the soil removed from the jar so that it could be cut into sections (L 315-316)?

Reviewer #2 (Remarks to the Author):

Response to Witzgall et al. 2021 Soil organic carbon under lockdown: Fresh plant litter as the nucleus for persistent carbon.

In this manuscript Witzgall et al. investigate the effects of labelled litter addition and a change in texture (coarse or fine) on microcosm experiments over a 95-day incubation. During the incubation the authors measured heterotrophic respiration and its ^{13}C abundance. Following the end of the incubation, the authors fractionated their microcosms into separate pools of organic matter using a combined density and size-based separation method. The authors then measured organic carbon and its isotopic composition with an EA-IRMS, its characteristics with ^{13}C CP-MAS NMR, the microbial community composition using phospholipid fatty acid analysis, and investigated the spatial arrangement / microscale distribution of the samples with SEM and NanoSIMS. The authors successfully demonstrate that fungi responded strongly to the introduced litter and that this corresponded, particularly in the coarse treatment, to increased respiration, transformation of organic matter, and its storage within the mineral associated, and more clearly, the occluded fraction. The study is well replicated and provides further strong-evidence of the link between microbial transformation processes and the storage of organic matter in soils (occluded and mineral-associated pools). Overall, I thoroughly enjoyed reading this insightful and generally well written manuscript.

This manuscript deserves to be published in a notable journal such as Nature Communications after some minor corrections. While the manuscript is generally well written, I would propose that the authors could still increase the clarity of its language in several sections, particularly the abstract, and with regards to their methods (experimental set-up / which specific samples were measured during each analysis / number of observations). The manuscript is also well referenced but could still benefit from a couple of prominent references in the discussion to help put their findings into context of the wider literature. Once these small details have been corrected, I would recommend that this manuscript is ready for publication. Overall, I thank the authors for their contribution and look forward to reading the final accepted manuscript.

I've included my more general comments under the specific subheadings of the manuscript, with other minor formatting or text corrections:

Abstract.

Overall, I felt that the abstract could be more specific, covering the methods, findings and conclusions in more detail. After reading it, I still didn't truly understand how you came to these conclusions, what you had measured, or the experimental design. I "briefly" revisited Nature communications' guidelines

for authors (so may have missed a set of criteria for the abstract) and while they state that the abstract should be written for a more general scientific audience, I found that your abstract didn't sell the true implications of your work. Please adjust it to be more specific.

Line 31: the soil matrix

Line 37: concurrently is more specific than in concert.

Line 39: on the surface of fresh litter, which acts as a key...

Introduction.

Line 46: Microbial growth and activity (ref).

Line 47: Is this supposed to be a separate paragraph? It seems as though there is a break in the narrative?

Line 54: Please reduce the size of this sentence.

Line 65: a systematic approach to investigating...

Methods

I have a couple of more general comments regarding your methods section, particularly regarding your experimental setup. I found that it wasn't very clear how you had obtained your specific experimental treatments. How much sand was added? Was the reason that you included 120 g for the coarser and 90 g of the finer soil in the separate microcosms to account for an absolute dilution in C content caused by the introduction of sand? Was the quartz sand double autoclaved or irradiated? before introduction to prevent any biological contamination of one treatment relative to another? Is this done by the supplier?

I commend your use of a conceptual model in the discussion to help explain your observations and found it pedagogic. As a second, more general comment, I do however feel that your experimental setup could benefit from a similar figure, to further explain the experimental design and measurements. If not included in the main manuscript, at least included as a supplementary figure? Detailing the specific treatments and what was measured on each treatment, fraction, and number of observations. This sort of framework would help a reader that is less accustomed to these methods.

Line 307: ") (" should be replaced with a ";" throughout.

Line 309: ". Carbonic acid with a known isotopic composition, diluted in helium, ..."

Line 325: How was the sonicator calibrated? A simple citation or two at the end of these brackets would suffice.

Line 329: rinse solution?

Line 346, 347, & 350: Was this 165 ppm or 160 ppm? Or both?

Line 354 and following equations: Is all of the text in the equations supposed to be in italics?

Line 375: The square and normal brackets likely need to be reversed

Results

My main general comment in the result section pertains to the lack of data and presentation of the SEM/NanoSIMS results in this work. I would have expected more images, text in the results, and detail covering this section as it was built-up extensively (the importance of looking at microstructure) through the manuscript (and looks interesting). Could you please include more detailed observations and results (SEM/NanoSIMS) in the manuscript or within the supplementary information?

Line 81-82: This sentence "We report the" doesn't seem like it belongs in the results section? Please remove it if it's unnecessary.

Line 90: Could you please use a different colour for the soil-derived heterotrophic respiration, brown maybe? It was currently difficult to see. Could it also be put under the litter derived contribution?

Discussion

The discussion would benefit from several prominent citations, related to your findings (and I'm very sorry if I missed some with the Vancouver citation style of Nature). Specifically, could you please reference some more of prominent C. Chenu / Oades' papers covering biological aggregation

processes in soils. Some more details about how your observations sit within the body of literature that has used NanoSIMS to evaluate SOC and its transformation processes in soils could also be warranted (Mueller / Schweizer et al.. etc...). I also believe that the decomposition continuum by Lehmann and Kleber (2015, 2019) could be cited with regards to your observations of microbial transformation processes contributing to the preservation and storage of organic matter in occluded and mineral-associated pools (like 246).

Line 232: Our microcosm experiments suggest that, in...

Line 251: promotes the cohesion...

Data availability

It's not necessary for publication, but could more data please be included in the supplementary information? I couldn't actually see a supplementary information section?

RESPONSE TO REVIEWERS

Thank you again for submitting your manuscript "Soil organic carbon under lockdown: Fresh plant litter as the nucleus for persistent carbon" to Nature Communications. We have now received reports from 2 reviewers and, on the basis of their comments, we have decided to invite a revision of your work for further consideration in our journal. Your revision should address all the points raised by our reviewers (see their reports below). In particular, the revised manuscript should better discuss the caveats and limitations from only using one soil type and one litter type (Reviewer #1). Further, Reviewer #2 felt that more NanoSIMS images should have been included, and stressed that the manuscript should be edited for clarity, in particular the Methods section.

Reviewer #1

The manuscript by Witzgall et al. examines how leaf litter is ultimately processed and is incorporated into soil organic matter. I found the research here to be very interesting and is the first to my knowledge to visually observe the process of SOM formation. I believe that this will be of interest to a wide range of microbial and soil ecologists.

We thank the reviewer for their constructive and insightful comments. Our detailed answers are listed below.

The major issue that I have with this manuscript is that only one soil and one litter type was used in this experiment. I believe that the authors need to discuss the potential caveats that this limitation might impose. For instance how might litter quality influence the observed processes? Could variation in the soil microbial community (i.e. different soils) influence this process? As it currently stands the conclusions from this study rest on the assumption that all soils will be similar to that of a single ag soil decomposing corn litter.

You are certainly raising an important question of the experimental setup, and we now address this in l. 321 and l. 335. We aimed at studying fundamental principles of SOM formation in a soil that is highly representative. Therefore, we specifically collected the soil for our experiment from an Ap horizon of a Cambisol as it is a common and typical agricultural soil. Maize was chosen as litter substrate as it is a crop grown worldwide in agricultural systems. Instead of focusing on elucidation of natural heterogeneity, we aim at providing a deeper functional understanding of the link between plant litter decomposition and SOM formation at a mechanistic level.

Additionally, a minor point, but how was the soil removed from the jar so that it could be cut into sections (L 315-316)?

Thank you for pointing this out! The microcosms were specifically designed for this experiment to allow for a precise separation of the three depth increments. They were made of two halves so that they could easily be opened up after incubation. To better illustrate the setup, we have now added photos of the microcosms to demonstrate how the depth increments were sampled (Supplementary Fig. 4). Furthermore, we have added a detailed description in the manuscript (l. 372).

Reviewer #2

Response to Witzgall et al. 2021 Soil organic carbon under lockdown: Fresh plant litter as the nucleus for persistent carbon. In this manuscript Witzgall et al. investigate the effects of labelled litter addition and a change in texture (coarse or fine) on microcosm experiments over a 95-day incubation. During the incubation the authors measured heterotrophic respiration and its ^{13}C abundance. Following the end of the incubation, the authors fractionated their microcosms into separate pools of organic matter using a combined density and size-based separation method. The authors then measured organic carbon and its isotopic composition with an EA-IRMS, its characteristics with ^{13}C CP-MAS NMR, the microbial community composition using phospholipid fatty acid analysis, and investigated the spatial arrangement / microscale distribution of the samples with SEM and NanoSIMS. The authors successfully demonstrate that fungi responded strongly to the introduced litter and that this corresponded, particularly in the coarse treatment, to increased respiration, transformation of organic matter, and its storage within the mineral associated, and more clearly, the occluded fraction. The study is well replicated and provides further strong-evidence of the link between microbial transformation processes and the storage of organic matter in soils (occluded and mineral-associated pools). Overall, I thoroughly enjoyed reading this insightful and generally well written manuscript. This manuscript deserves to be published in a notable journal such as Nature Communications after some minor corrections.

While the manuscript is generally well written, I would propose that the authors could still increase the clarity of its language in several sections, particularly the abstract, and with regards to their methods (experimental set-up / which specific samples were measured during each analysis / number of observations). The manuscript is also well referenced but could still benefit from a couple of prominent references in the discussion to help put their findings into context of the wider literature. Once these small details have been corrected, I would recommend that this manuscript is ready for publication. Overall, I thank the authors for their contribution and look forward to reading the final accepted manuscript.

We are thankful for the comprehensive and encouraging evaluation of our manuscript. We would also like to acknowledge your work and dedication invested—it helped us to thoroughly improve the quality of our manuscript. Please find our detailed answers to your comments below.

I've included my more general comments under the specific subheadings of the manuscript, with other minor formatting or text corrections:

Abstract.

Overall, I felt that the abstract could be more specific, covering the methods, findings and conclusions in more detail. After reading it, I still didn't truly understand how you came to these conclusions, what you had measured, or the experimental design. I "briefly" revisited Nature communications' guidelines for authors (so may have missed a set of criteria for the abstract) and while they state that the abstract should be written for a more general scientific audience, I found that your abstract didn't sell the true implications of your work. Please adjust it to be more specific.

Thank you for pointing us to this shortcoming. We have revised the abstract to provide more clarity for the reader. Accordingly, we have added some more specific information around our methods (l. 33) as well as our findings and conclusions (l. 42). Given the limitations by the word count (≤ 150), we hope that this change can be formally accepted.

Line 31: the soil matrix

This change has been implemented (l. 31).

Line 37: concurrently is more specific than in concert.

This change has been implemented (l. 39).

Line 39: on the surface of fresh litter, which acts as a key...

To reduce the word count, this sentence has been slightly modified.

Introduction

Line 46: Microbial growth and activity (ref).
References have been added (l. 52).

Line 47: Is this supposed to be a separate paragraph? It seems as though there is a break in the narrative?
The text has been separated into two paragraphs (l. 53).

Line 54: Please reduce the size of this sentence.
The sentence has been shortened (l. 60).

Line 65: a systematic approach to investigating...
This sentence has been modified (l. 71).

Methods

I have a couple of more general comments regarding your methods section, particularly regarding your experimental setup. I found that it wasn't very clear how you had obtained your specific experimental treatments. How much sand was added?

Thank you for this remark, we have reworked the explanation in the text (l. 328) and also added an illustration (Fig. 9). Sand was added before-hand to the initial bulk soil. While the finer-textured soil only contained the initial soil material (14% sand), we mixed 700.0 g of fresh field soil (equivalent to 583.1 g dry field soil) with 822.5 g artificial quartz sand. The mixture was then carefully homogenized and the texture was measured and confirmed to 60% sand content (Supplementary Table 1).

Was the reason that you included 120 g for the coarser and 90 g of the finer soil in the separate microcosms to account for an absolute dilution in C content caused by the introduction of sand? treatments. How much sand was added?

As we aimed to work at the same bulk density in all treatments, we added different amounts of the coarser- and finer-textured soil to the microcosms. We have clarified this in the manuscript (l. 328) and we further added Fig. 9 to support a better understanding of the experimental setting, sampling and analyses

Was the quartz sand double autoclaved or irradiated? before introduction to prevent any biological contamination of one treatment relative to another? Is this done by the supplier?

The quartz sand had been washed and air-dried by the supplier. We have added this information in l. 327. We assume the microbial content of the quartz sand to be negligible, which we conclude from identical PLFA based microbial community patterns between our control soils with and without quartz sand added. Hence, we think it is safe to exclude any contamination effects that could alter the scope of this study.

I commend your use of a conceptual model in the discussion to help explain your observations and found it pedagogic. As a second, more general comment, I do however feel that your experimental setup could benefit from a similar figure, to further explain the experimental design and measurements. If not included in the main manuscript, at least included as a supplementary figure? Detailing the specific treatments and what was measured on each treatment, fraction, and number of observations. This sort of framework would help a reader that is less accustomed to these methods.

We think that this is a great idea and agree that it would increase the understanding of our experimental set up. We have added a figure (Fig. 9) that captures sample preparation, experimental setup as well as an overview over the analyses carried out on which samples/number of replicates.

Line 307: “) (“ should be replaced with a “;” throughout.
This has been changed throughout the manuscript.

Line 309: “. Carbonic acid with a known isotopic composition, diluted in helium, ...”

This sentence has been modified (l. 366).

Line 325: How was the sonicator calibrated? A simple citation or two at the end of these brackets would suffice.

The sonicator was calibrated according to North 1976. The citation has been added accordingly (l. 382).

Line 329: rinse solution?

This change has not been implemented as “rinse solution” does not apply in our case.

Line 346, 347, & 350: Was this 165 ppm or 160 ppm? Or both?

The chemical shift region 110-160 ppm was applied when integrating the spectra (according to Mueller & Kögel-Knabner 2009). The regions mentioned later are those used in the molecular mixing model (according to Nelsan & Baldock 2005).

Line 354 and following equations: Is all of the text in the equations supposed to be in italics?

We have changed it to regular font throughout the manuscript.

Line 375: The square and normal brackets likely need to be reversed

This change has not been implemented as we think that the square and normal brackets are correct (see e.g. Baumert et al., 2018).

Results

My main general comment in the result section pertains to the lack of data and presentation of the SEM/NanoSIMS results in this work. I would have expected more images, text in the results, and detail covering this section as it was built-up extensively (the importance of looking at microstructure) through the manuscript (and looks interesting). Could you please include more detailed observations and results (SEM/NanoSIMS) in the manuscript or within the supplementary information?

Thank you for highlighting this. We agree that the manuscript would benefit from more results presented from the SEM/NanoSIMS imaging. Detailed information and SEM micrographs has been added in the result section (l. 204-207 and Fig. 6) and we further present additional data and images in supplements (Supplementary Fig. 1-3).

Line 81-82: This sentence “We report the” doesn’t seem like it belongs in the results section? Please remove it if it’s unnecessary.

This makes sense. While we are aware that it might be redundant, we believe that for some readers it could be important to acknowledge that this data is in fact normalized. To ensure that the descriptive text flows better, we have instead moved the sentence to the figure caption (Fig. 1).

Line 90: Could you please use a different colour for the soil-derived heterotrophic respiration, brown maybe? It was currently difficult to see. Could it also be put under the litter derived contribution?

We have adjusted the colour to improve the visibility (Fig. 1). However, we are not able to change the order of the displayed data. We are not displaying stacked values, but instead the absolute cumulative values as highlighted in the bar charts (Fig 1. b and d). As the soil-derived CO₂ is higher than litter-derived CO₂, the order can therefore not be changed.

Discussion

The discussion would benefit from several prominent citations, related to your findings (and I’m very sorry if I missed some with the Vancouver citation style of Nature). Specifically, could you please reference some more of prominent C. Chenu / Oades’ papers covering biological aggregation processes in soils. Some more details about how your observations sit within the body of literature that has used

NanoSIMS to evaluate SOC and its transformation processes in soils could also be warranted (Mueller / Schweizer et al.. etc...). I also believe that the decomposition continuum by Lehmann and Kleber (2015, 2019) could be cited with regards to your observations of microbial transformation processes contributing to the preservation and storage of organic matter in occluded and mineral-associated pools (like 246).

Thank you for these suggestions. We have now underlined our findings with relevant literature on l. 273, l. 281, l.287 and l. 289.

Line 232: Our microcosm experiments suggest that, in...

This change has been implemented (l. 265).

Line 251: promotes the cohesion...

This change has been implemented (l. 287).

Data availability

It's not necessary for publication, but could more data please be included in the supplementary information? I couldn't actually see a supplementary information section?

We have added a substantial amount of data in supplements (Supplementary Fig. 1-4, Supplementary Table 1-4).

REVIEWERS' COMMENTS

Reviewer #1 (Remarks to the Author):

The authors have done a good job addressing my comments. One additional detail would be welcome. Could the authors please provide details regarding where/how the polymethylene microcosms were sourced?

Reviewer #2 (Remarks to the Author):

Great work, thanks for considering the suggestions.

This article is ready for publication.

Kind regards,

Reviewer 2

RESPONSE TO REVIEWERS

We would like to thank both reviewers for their encouraging words. Please find our answer below to the remark of Reviewer #1.

Reviewer #1 (Remarks to the Author):

The authors have done a good job addressing my comments. One additional detail would be welcome. Could the authors please provide details regarding where/how the polymethylene microcosms were sourced?

The material for the microcosms, 1.4 g cm^{-3} polyoxymethylene, was sourced from Sahlberg GmbH in Munich, Germany. These details have now been added in the manuscript (see I. 337). The microcosms were then constructed so that they could be opened up after incubation (as described more in detail previously).

Reviewer #2 (Remarks to the Author):

Great work, thanks for considering the suggestions.

This article is ready for publication.

Kind regards,

Reviewer 2